# Fatigue-Induced Inter-Limb Asymmetries in Strength of the Hip Stabilizers, Postural Control and Gait Following a Unilateral Countermovement Vertical Jump Protocol

**DOI:** 10.3390/sports9030033

**Published:** 2021-02-27

**Authors:** Ioannis Konstantopoulos, Ioannis Kafetzakis, Vasileios Chatziilias, Dimitris Mandalidis

**Affiliations:** Sports Physical Therapy Laboratory, Department of Physical Education and Sports Science, School of Physical Education and Sports Science, National and Kapodistrian University of Athens, 172 37 Athens, Greece; konstantopoulos.giannos@gmail.com (I.K.); kafetzos13@icloud.com (I.K.); vasilischatz@phed.uoa.gr (V.C.)

**Keywords:** lower limb, proximal stability, overloading, motor control, injury prevention

## Abstract

It is generally accepted that neuromuscular overload and fatigue of one lower limb can affect the functional ability of the ipsilateral limb, and possibly the contralateral limb, increasing the likelihood of injury. The purpose of the current study was to examine the effect of a unilateral countermovement vertical jump (UCVJ) fatigue protocol on the neuromuscular function of the ipsilateral as well as the contralateral lower limb. The isometric strength of the hip stabilizers, postural control via posturographic analysis during the Y-Balance-Test (YBT), and the stance-phase-of-gait were assessed in 24 healthy physical active males and females before and after execution of a UCVJ fatigue protocol. The fatigue protocol included 5 sets of 20 maximum UCVJs performed on the supportive leg, with a 30-s break between sets. Following a 16.8% decline in vertical jump performance and an associated 2.3-fold increase in perceived exertion, our findings revealed significant post-fatigue inter-limb differences regarding postural control. The post-fatigue inter-limb differences regarding the isometric strength of the hip stabilizers and the stance-phase-of-gait parameters were not significant. Our findings showed that a 100 UCVJs session is likely to induce significant inter-limb differences in postural control, possibly increasing the risk of lower limb injury.

## 1. Introduction

Inter-limb asymmetries in sports, i.e., the differences between the limbs in terms of their functional ability (e.g., flexibility, range of motion strength and motor control) due to the predominant use of one limb over the other [1], are particularly common in athletes involved in unilateral as well as bilateral sporting activities [1,2,3]. Such asymmetries have received a lot of attention from sport coaches and clinicians as they have been associated with both a decrease in athletic performance [4,5] and an increase in the risk of non-contact injuries [6]. In the context of non-contact injury prevention, several authors have also investigated inter-limb asymmetries under the influence of exercise-induced fatigue [7,8,9,10,11]. With fatigue being an inevitable consequence of athletic performance, it has been argued that it can either exacerbate pre-existing differences between the limbs or induce them if they are non-apparent. [6]. However, despite the growing interest in this issue, the number of studies conducted is limited and the available findings remain unclear. This is primarily due to the variety of study designs (testing pre/post or during the fatigue protocol) [7,10,11] and the activity or task (e.g., running, race walking, jumping, squatting, soccer, rowing) implemented to induce fatigue [7,8,9,10,11]. Furthermore, the parameters used to assess inter-limb asymmetry (e.g., squats, single-leg countermovement jumps, gait analysis, or isokinetic strength testing) have covered only certain aspects of lower limbs function [6], leaving room for further and more extensive investigation of this issue.

The effect of fatigue on lower limb function has also been examined in terms of dynamic hip joint stability, primarily due to its potential association with non-contact injuries such as ACL rupture [12], anterior knee or patellofemoral joint pain [13] and inversion ankle sprains [14]. Clinically, these injuries can be justified by the fact that muscle dysfunction of proximally located lower limb joints, such as the hip joint, may require compensation of a more distal joint that is linked in series within the same kinetic chain, compromising the function of the entire lower limb [15]. Experimentally, it has been shown that fatigue-induced muscle weakness of certain hip stabilizers (e.g., hip abductors) can compromise the function of the ipsilateral lower limb by altering the unipedal lower limb kinematics (e.g., medial knee displacement) [16,17], postural control [18,19], and hip position sense [20], as well as the gait parameters [19,20]. Other authors using a more “functional” fatigue protocol on lower limb muscles such as unilateral semi squats and a combination of single-limb plyometric drills and squats revealed impaired postural control [18,21] and a lack of absorbing the ground reaction forces following landing on the ipsilateral lower limb [22].

Although the amount of available data regarding lower limb dysfunction due to fatigue of the hip stabilizers is overwhelming, they are limited to the effects of a unilateral fatigue protocol only on the ipsilateral lower limb with exercises that are not as functional as those performed in sports. Consequently, it remains unclear whether a unilaterally performed fatigue protocol that requires the execution of a more sport-related exercise, such as a unilateral vertical jump, can potentially affect the function of the ipsilateral, as well as the contralateral lower limb inducing an inter-limb asymmetry to such an extent that it may increase the likelihood of injury. Jumping and landing on one foot are key components of many ball games such as basketball and team handball with athletes performing as many as 100 jumps per game/event [23,24] and probably twice as many in consecutive training sessions. The execution of these jumps, which is required in these team sports as well as in individual sports such as artistic and rhythmic gymnastics and figure skating, increase the risk of injury to these athletes, especially during the landing phase following jumping [25,26] on a small support base while absorbing the vibrations created by the ground reaction forces. 

Knowing the potential effects of fatigue on the neuromuscular function of the lower limbs will enable coaches and sporting instructors to decide on the exposure of trainees to the training loads. In addition, health care providers (e.g., physical therapist, athletic trainers) will be able to design and implement appropriate exercise programs to prevent or to restore certain aspects of lower limb function. Therefore, the purpose of this study was to investigate the acute effects of a unilateral countermovement vertical jump (UCVJ) protocol on the bilateral function of the lower limbs in terms of hip stabilizers’ muscle strength, the posturographic assessment of postural control, and the spatio-temporal and force-related evaluation of the stance-phase-of-gait.

## 2. Materials and Methods

### 2.1. Participants

Twenty-four healthy, physically active collegiate students (12 males and 12 females, mean ± SD age of 22.0 ± 2.0 y, height: 1.7 ± 0.1 m, body weight: 67.7 ± 12.3 kg and BMI: 22.9 ± 2.7 kg·m^−1^) from the local University School of Physical Education and Sports Science, with no systematic involvement in bilateral or unilateral sporting activities, volunteered to participate in the study. These individuals were selected because they were less likely to show differences between the lower limbs for the parameters measured, as they were not expected to have developed specific athletic skills due to regular involvement in a particular sport. Participants with pain or inability to fully weight-bear and walk without limping for at least 3 months before participation to this study, a history of significant injury or surgery to the lower limb or spine, neurological, visual, vestibular, or balance disorders, and medication that may affect the measurements (e.g., performance enhancing substances) were excluded from the study. Individuals with obvious musculoskeletal abnormalities such as leg length inequality, excessive scoliosis and foot deviations (e.g., overpronation or supination) were also excluded following a standard evaluation of the musculoskeletal system. The study protocol was approved by the University’s Human Research Ethics Committee (Registration No. 1175/11-03-2020) and each participant signed an informed written consent prior to testing.

### 2.2. Measures

#### 2.2.1. Isometric Strength of the Hip Stabilizers 

The strength of the hip adductors/abductors and internal/external rotators muscles was measured with isometric maximal voluntary contractions (IMVC) using an S-type digital force-gauge for tension and compression measurements, with an acquisition frequency of 2000 Hz (FH-5K, Sauter GmbH, Balingen, Germany), modified to perform hand-held measurements. For that purpose, the force gauge was connected in series with a cylindrical aluminum tube and two curved aluminum plates covered with soft rubber material, which were fixed to the free ends of the force gauge and the tube. The plate attached to the free end of the force gauge was placed against the examinee’s body part that was under investigation while the plate attached to the free end of the tube was placed on the examiner’s body, against which force was applied.

Strength measurements of the abductor and adductor muscles of the hip were performed by placing the attached plate on the force gauge 5-cm proximal to the lateral and medial femoral condyle, respectively, with each participant supine on the treatment table and the knee of the limb under investigation fully extended [27]. The contralateral limb was on the treatment table either with the knee fully extended or hung over the side of the table with the knee flexed at 90° for the hip abductors and adductors strength measurement, respectively. The strength of the hip external and internal rotators was measured by placing the plate attached to the force gauge 5-cm proximal to the medial and lateral malleolus, respectively, with each subject seated in a test chair and stabilized with a strap around the waist, the hip in neutral rotation and the knees at 90° of flexion [28]. Both testing procedures have shown acceptable reliability (ICC: 0.84–0.98) for clinical and research purposes [28,29]. The participants were instructed to stabilize themselves by holding on to the sides of the table with their hands but not to exert any force during testing. Each participant performed two 5-sec sub-maximal isometric contractions at 50% and 75% of the MVC for familiarization followed by three 5-sec maximal isometric contractions with a 1-min rest between repetitions. The average of three valid IMVCs was used in data analysis.

#### 2.2.2. Assessment of Postural Control

The posturographic-based assessment of postural control was performed with each participant reaching with one leg in the anterior (A), posteromedial (PM), and posterolateral (PL) directions of the Y-Balance Test (YBT), while standing barefoot with the opposite leg on a plantar pressure distribution platform (FDM-S Measuring System for Force Distribution, Zebris Medical GmbH, Isny im Allgäu, Germany) [30]. The YBT has shown adequate reliability in measuring dynamic postural control (ICC: 0.85–1.00) [30]. The foot pressure signals were recorded at a sampling rate of 120 Hz and analyzed with the WinFDMS computer software (WinFDMS v.0.1 for Windows, Zebris Medical GmbH, Isny im Allgäu, Germany). Each participant was asked to perform three practice trials with each limb and three more attempts for the actual test in each one of the three directions. The average distance expressed as a percentage of the length of the non-supportive lower limb and the corresponding average velocity (VL), and the anteroposterior and mediolateral displacement (APd and MLd) of the center of pressure (CoP) of the three successful test attempts in each direction were used in data analysis.

#### 2.2.3. Stance-Phase-of-Gait Assessment

The stance-phase-of-gait was assessed in terms of the spatio-temporal and force-related parameters listed in Table 1 with each participant stepping barefoot on the aforementioned plantar pressure distribution platform as he/she was walking with the average human walking rate of 100 steps per minute using a metronome. All participants were instructed to look straight ahead and step on the platform without adjusting their stride having walked 3 steps before and another 3 steps after contacting the platform. A sufficient number of trials were given to allow subjects’ familiarization with the process of walking over the platform and to ensure reliable within-session data. The mean of 20 sequences of steps (10 with the right and 10 with the left) performed by each participant was included in the analysis. Stance-phase-of-gait-analysis has been found to be a valid and reliable measurement in the healthy population [31].

### 2.3. Design and Procedure

All subjects were referred to the laboratory for two consecutive visits, with an interval of 2–5 days between visits. At the first visit, all participants provided information about their medical history and underwent anthropometric measurements as well as musculoskeletal assessment to determine their eligibility for the study. Once eligibility was established, each participant was assessed on isometric strength of the hip abductors/adductors and the internal/external rotators, posturographic-based postural control during the execution of the YBT, and the stance-phase-of-gait. A standard warm-up procedure that included 5 min of cycling using a stationary bike and 5 min of dynamic lower limb muscle stretching was performed before the tests. The same measurements were repeated in the second visit, following the execution of 100 maximum countermovement vertical jumps (5 bouts of 20 repetitions each with a 30-sec break between the bouts) in each participant’s supportive lower limb. The supportive limb (16 right and 8 left) was determined based on the Waterloo Footedness Questionnaire—Revised [32]. All participants were advised to maintain their physical activity habits between the testing sessions.

The UCVJ fatigue protocol by means of the height of the vertical jumps and the corresponding flight time was monitored using an inertial measurement device (Gyko, Microgate S.r.l., Bolzano, Italy) with an acquisition frequency of 1000 Hz, placed on each participant’s waist using an adjustable belt. The perceived exertion during UCVJ was also rated with the Borg’s 15-point scale [33].

### 2.4. Statistical Analysis

The sample size (n = 24) with which statistical significance could be achieved with a = 0.05, 80% power and effect size (f) = 0.2526 (based on a partial η^2^ = 0.06) was calculated a priori. The effects of the UCVJ fatigue protocol on the vertical jump-related parameters and the perceived exertion were examined with one-way repeated measures ANOVA. The effect of the UCVJ fatigue protocol on both the fatigued and the non-fatigued lower limb with regard to strength of the hip stabilizers, the posturographic-based assessment of postural control and the stance-phase-of-gait related parameters were examined using two-way ANOVA for repeated measures. Significant main effects were followed by pairwise comparisons using Bonferroni adjustment. Statistical analyses were conducted in SPSS, version 25.0 (IBMCorp, Armonk, NY, USA).

## 3. Results

The unilateral vertical jump training protocol yielded a significant reduction in the average vertical jump height (−16.9%, *p* < 0.001, η^2^ = 0.46) and the flight time (−8.8%, *p* < 0.001, η^2^ = 0.39). These changes were associated with a significant increase in the perceived exertion measured with the Borg’s 15-point scale (124%, *p* < 0.001, η^2^ = 0.46).

### 3.1. Effect of the UCVJ on the Isometric Strength of the Hip Stabilizers

The results of the study showed that the differences between the lower limbs in both the pre- and post-fatigue condition and between conditions for the same lower limb regarding the maximum isometric strength of the hip stabilizers and the agonist/antagonist hip strength ratios were not significant (Table 2).

### 3.2. Effect of the UCVJ on Postural Control during the Y-Balance Test

The distances reached while standing on the fatigued lower limb were non-significantly decreased in the majority of directions (AN and PM) following the UCVJ protocol. In contrast, the distances reached while standing on the non-fatigued lower limb remained either unchanged (PM) or increased (AN and PL) with a significant limb-by-time interaction being found for the distance reached only in the PL direction (*p* = 0.039, η^2^ = 0.17). Post hoc analysis revealed a significant post-fatigue inter-limb difference regarding the distances reached in both the AN (*p* = 0.024) and PL direction (*p* = 0.033, Table 3).

Statistical analysis yielded a significant limb-by-time interaction also for the VL of CoP (*p* < 0.001, η^2^ = 0.39) during reaching in the anterior direction. Post hoc analysis revealed a significant increase of the VL of CoP in the fatigued (*p* = 0.025) and a non-significant decrease of the VL of CoP in the non-fatigued limb following the UCVJ protocol. These changes resulted in a significant inter-limb difference regarding the VL of CoP in the post-fatigue condition (*p* < 0.001). Similarly, the non-significant increase and the non-significant decrease in the MLd of CoP of the fatigued and non-fatigued lower limb, respectively, resulted in a significant post-fatigue inter-limb difference during reaching in the AN direction (*p* = 0.026). The inter-limb differences in the APd of CoP were not significant either before or after the implementation of the UCVJ fatigued protocol (Table 3).

The inter-limb differences regarding the VL, the MLd and the APd of CoP were not significant during reaching in the PM and PL directions, either before or after the execution of the UCVJ protocol. An exception was the APd of CoP during reaching in the PM direction, which demonstrated a significant reduction in the non-fatigued lower limb following the fatigue protocol (*p* = 0.023, Table 3).

### 3.3. Effect of the UCVJ on the Stance-Phase-of-Gait

The differences between the fatigued and the non-fatigued lower limbs following the UCVJ protocol were non-significant for all the stance-phase-of-gait related parameters measured (Table 4).

## 4. Discussion

For a constructive discussion, the results of the present study should be viewed in the context of whether sufficient muscle fatigue was induced by an exercise protocol implicating the joints of the entire lower limb. The fatigue induced by this protocol was determined based on a 16.9% decline in the vertical jump height, which corresponded to a 8.8% reduction in flight time, as well as an almost 2.3-fold increase in perceived exertion measured by Borg’s 15-point RPE scale (7.5 points before vs. 16.8 points after the UCVJ fatigue protocol). Similar levels of fatigue have been reported in previous studies, some of which consider the point of fatigue as the reduction of the height of one- or two-legs jumps by 10%–20% of the baseline jump [34,35], and others when the perceived exertion approximated 17 out of 20 points on the Borg’s 15-point RPE scale [36,37].

Based on the fatigue criteria that were ultimately met by the UCVJ protocol, it was revealed that posturographic-based measurements, mainly by means of the VL and the MLd of the CoP, were increased on the FLL and decreased on the NFLL primarily during reaching in the anterior direction of the YBT, resulting in post-fatigue inter-limb differences in postural control. These changes were also reflected in differences between the two extremities regarding the distances reached in two of the three directions of the YBT. Similar findings have been reported in a previous study in which the stability of the body was disturbed by a plyometric exercise protocol designed to elicit symptoms of muscle damage by performing 200 vertical jumps [38]. Other authors also reported balance disturbances following plyometric exercises without reporting whether participants were fatigued or not. Romero-Franco and Jiménez-Reyes [39] showed increased length and velocity of center-of-pressure movement in the right leg support stance compared with the baseline recordings and the control group immediately after a plyometric exercise protocol. Despite a slight recovery, these parameters remained deteriorated 5 min later. In a later study, the authors found meaningful but not significant changes in unilateral balance performance following vertically performed plyometric exercises in youth soccer players [40].

It is generally accepted that the increased VL of CoP is a negative outcome associated with falls in the elderly [41] and with neurological problems such as Parkinson’s disease [42]. In uninjured athletes, the VL of CoP is lower than in controls, indicating a positive effect of athletic skill on this postural control factor [43]. However, this factor is increased after an exercise-induced fatigue protocol in the athletic population, indicating a negative outcome [44]. The changes in postural control achieved during YBT could be partly explained by the corresponding reduction in strength of some of the most important hip stabilizers, such as the hip abductors and rotators, which also play an important role in the execution of this test [45]. However, the maximum isometric strength of these muscles not only did not decrease to the extent that could affect postural control (25%–30%) [44], but there was a small increase in strength after the UCVJ fatigue protocol. Our evidence suggests that the hip stabilizers were not fatigued enough, or at least in a manner similar to the one used in previous studies, to show a decrease in their isometric strength after the UCVJ fatigue protocol [17,18,20,46]. The majority of these studies showed significant reductions in the strength of the hip stabilizers, but these changes were achieved primarily after performing a fatigue protocol that targeted the specific musculature.

There are several factors that may be involved in the inter-limb differences regarding post-fatigue postural control, most of which are related to changes that occur in the muscles that play a leading role in vertical jump performance and are metabolic and/or sensory in nature. Paillard [44], after extensively reviewing the effects of local fatigue on postural control reported that its degradation is inevitable after voluntarily fatiguing several muscles, particularly of the lower limbs such as the ankle plantar flexors and knee extensors, and that this phenomenon is accentuated when body balance is maintained in the unipedal stance. Apparently, the protocol used in the present study to induce fatigue met all of the above criteria as postural control appeared to degrade at least during execution of YBT, as the lower limb muscles became fatigued with repetitive vertical jumps. Evidently, the deterioration of postural control in the fatigued lower limb is more likely to be associated with alterations in neuromuscular transmission, muscle action potential propagation, excitation–contraction coupling and related contractile mechanisms, which are peripherally-related fatigue phenomena occurring in the muscle, the nerve endings and at the neuromuscular junction [47]. The fact that postural control changes were detected mainly by means of an increased VL of CoP could be due to an increase in the amplitude of physiological tremor. This condition is characterized by the low-amplitude rhythmic oscillations of a body part that occur around a fixed point when trying to maintain a steady posture or movement [48]. Although, changes in the amplitude of physiological tremor was not assessed in the present study, it can be hypothesized that the obtained increase of VL of CoP was the result of alterations in the sensitivity of the muscle spindles, the recruitment of large motor neurons and the low frequency fatigue that often follow eccentric muscle actions [49]. Furthermore, the changes in postural control were more easily detectable with reaching in the anterior direction, statistically speaking, probably due to the inherently lower stability exhibited by the supporting limb while reaching with the contralateral limb in this direction compared to the other directions [50].

The post-fatigued inter-limb differences in postural control were in part the result of the non-significant decrease in the VL of CoP of the non-fatigued limb. Previous studies have revealed that postural control while standing on one limb is adversely affected by fatiguing the opposite limb, but this depends on the duration and intensity of the fatigue protocol [51,52]. Indeed, when fatigue on one limb was induced centrally, with lower-intensity and longer-duration exercise, a greater disturbance in postural control of the non-fatigued limb developed [52], as opposed to the fatigue caused peripherally with short-term and higher-intensity exercise. It has been proposed that since cross-education depends on centrally-mediated mechanisms rather than peripheral, it is more logical to expect a negative effect on the postural control of the contralateral limb with a more centrally-targeted type of exercise as opposed to a type of exercise that aims to induce peripheral fatigue alone, such as the UCVJ fatigue protocol used in the present study.

Finally, the inter-limb differences for both spatio-temporal and strength-related parameters recorded during the support-phase-of-gait were not significant following performance of the UCVJ fatigue protocol. Previous studies have shown a significant increase in gait parameters, such as the single support time, stride time variability and step-to-step asymmetry in the frontal plane, as well as significantly slower mediolateral trunk movement in fatigued leg late stance toward the non-fatigued leg during fatiguing gait [19,20]. However, these changes have been observed only after locally fatiguing the abductor muscles of the hip [19,20]. Failure to find differences in the present study between the limbs in gait parameters may be due to the reorganization of multi-joint coordination and the redistribution of active muscle activity in such a way as to compensate for muscle dysfunction caused by muscle fatigue [53]. This phenomenon may be due to the integration of different sensory information from the central nervous system.

The extrapolation of the current findings to other populations should be performed with caution, as the number of UCVJs executed in the present study to induce fatigue was fixed (n = 100). Dancers, for example may be more resistant to fatigue as they perform approximately 200 jumps in 90 min during the day in technique classes, rehearsals and/or performances [51]. The method used to measure strength of the hip stabilizers may also have hidden possible muscle fatigue that was induced by the UCVJ protocol as isometric strength measurements have been poorly related with the fatigability of the hip abductors [54]. With hip stabilizers acting also eccentrically to decelerate the tilt of the pelvis towards to contralateral side during landing it would also be possible to detect strength deficits more easily if their strength was measured with this type of contraction. Furthermore, inter-limb differences regarding the stance-phase-of-gait parameters might be different if evaluated based on a self-selected walking pace as neuromuscular fatigue, at the end of an intense exercise protocol, is expected to have a different effect on trainees.

## 5. Conclusions

The results of this study showed that unilateral lower limb fatigue by means of 100 vertical jumps may increase inter-limb asymmetry, at least in terms of postural control. With inter-limb asymmetries being considered a predisposing factor for sports injuries, our findings show that such differences can emerge due to functional muscle fatigue, partly confirming the higher risk of injury that exists at the end of a game/training session where a similar number of jumps are expected to have taken place. These changes, however, cannot be explained by the fatigue of the hip stabilizers, as this did not occur from the fatigue protocol applied in the present study. Nevertheless, since fatigue is task-specific, the potential effect of an exercise-induced fatigue protocol on inter-limb asymmetries should be investigated in future studies by implementing different tasks and/or in populations with different athletic backgrounds.

## Figures and Tables

**Table 1 sports-09-00033-t001:** Definition of the parameters used in the assessment of stance-phase-of-gait.

Parameter	Definition
Gait line length	The perpendicular length of the line connecting the position of successive centers of pressure (CoP) calculated for successive moments in time during a single stance phase.
Contact time	The average contact time of the toes, mid-foot and heel, in seconds (s) and as a percentage of the total stance time.
Time change from heel to forefoot	The time elapsed between the points in time where the maximum force recorded on the heel and the forefoot, in seconds (s) and as a percentage (%) of the total stance time.
Maximum force and pressure	The average maximum values of force and pressure obtained for the toes, mid-foot and heel in N and N/cm², respectively.

**Table 2 sports-09-00033-t002:** Means and standard deviations (in brackets) of the maximum isometric strength of the hip stabilizers and the agonists/antagonists isometric strength ratios before and after the execution of the unilateral countermovement vertical jump (UCVJ) fatigue protocol.

	Pre-Fatigue		Post-Fatigue	
	FLL	NFLL	FLL	NFLL
HABD (N)	218.9 (45.0)	214.1 (55.1)	226.9 (54.6)	226.3 (56.7)
HADD (N)	323.8 (91.1)	306.5 (64.8)	350.6 (90.3)	347.9 (83.7)
HEXR (N)	146.4 (43.9)	143.3 (36.8)	148.4 (37.8)	144.7 (33.3)
HINR (N)	178.5 (55.9)	165.9 (46.7)	178.2 (45.0)	182.9 (52.7)
HABD/HADD	0.7 (0.1)	0.7 (0.2)	0.7 (0.1)	0.7 (0.1)
HEXR/HINR	0.9 (0.2)	0.9 (0.3)	0.9 (0.3)	0.8 (0.2)

NOTE: FLL = Fatigued lower limb; NFLL = Non-fatigued lower limb; HABD = Hip abductors; HADD = Hip adductors; HEXR = Hip external rotators; HINR = Hip internal rotators.

**Table 3 sports-09-00033-t003:** Means and standard deviations (in brackets) of the posturographic parameters recorded during reaching in the three directions of the Y-Balance Test (YBT) with both lower limbs before and after the execution of the UCVJ fatigue protocol.

Postural Control Parameters	YBT Directions	Pre-Fatigue		Post-Fatigue	
		FLL	NFLL	FLL	NFLL
RD (cm)	AN	83.9 (5.6)	83.7 (6.4)	82.7 (5.3)	84.1 (5.3) *
	PM	105.8 (7.8)	106.1 (7.5)	105.1 (7.5)	106.0 (7.6)
	PL	100.1 (8.6)	99.9 (8.7)	100.7 (6.7)	103.1 (7.9) * ^†^
VL of CoP (mm/sec)	AN	50.7 (9.6)	51.9 (11.0)	55.1 (12.1) ^§^	48.7 (8.7) **
	PM	54.5 (9.8)	53.1 (9.6)	56.2 (10.7)	53.1 (10.3)
	PL	58.4 (9.9)	57.2 (8.4)	58.2 (9.7)	55.1 (8.8)
MLd of CoP (mm)	AN	12.6 (2.3)	12.8 (2.5)	13.3 (2.7)	12.1 (1.9) *
	PM	12.5 (3.1)	12.4 (2.7)	12.6 (3.2)	12.3 (2.5)
	PL	10.8 (3.4)	11.3 (2.7)	11.7 (3.6)	12.2 (2.7)
APd of CoP (mm)	AN	44.9 (12.7)	43.2 (14.0)	43.6 (12.1)	46.4 (14.1)
	PM	22.7 (8.0)	23.6 (8.8)	22.7 (8.7)	19.0 (5.5) ^‡^
	PL	17.2 (6.8)	16.8 (5.6)	18.6 (8.0)	18.6 (5.6)

Note: FLL = Fatigued lower limb; NFLL = Non-fatigued lower limb; AN = Anterior direction; PM = Posteromedial direction; PL = Posterolateral direction; RD = Reaching distance; CoP = Center of pressure; VL = Velocity; MLd = Mediolateral displacement; APd = Anteroposterior displacement; * *p* < 0.05 and ** *p* < 0.001 between the FLL and NFLL at the post-fatigue condition. ^†^
*p* < 0.05 and ^‡^
*p* < 0.01 between the pre- and post-fatigue condition for the NFLL. ^§^
*p* < 0.05 between the pre- and post-fatigue condition for the FLL.

**Table 4 sports-09-00033-t004:** Means and standard deviations (in brackets) of temporal and load distribution characteristics for both lower limbs before and after the execution of the UCVJ fatigue protocol.

Gait Parameters	Foot Zone	Pre-Fatigue		Post-Fatigue	
		FLL	NFLL	FLL	NFLL
GL (mm)		237.3 (18.9)	238.5 (18.4)	237.8 (20.9)	239.5 (20.4)
T_c_ (s)		0.73 (0.02)	0.73 (0.02)	0.73 (0.03)	0.74 (0.03)
T_c_ (%)	HEE	19.1 (3.7)	19.7 (4.8)	19.0 (3.5)	18.9 (2.2)
	MID	41.4 (9.3)	42.3 (9.2)	43.2 (8.2)	45.2 (7.9)
	FOR	75.9 (1.4)	75.3 (1.4)	75.7 (1.5)	75.5 (1.2)
T_h–f_ (s)		0.26 (0.06)	0.27 (0.04)	0.27 (0.04)	0.29 (0.04)
T_h–f_ (%)		35.9 (7.7)	37.3 (5.3)	37.5 (5.3)	38.8 (4.5)
F_max_ (N)	HEE	486.9 (89.7)	597.4 (84.4)	487.0 (85.7)	511.1 (86.1)
	MID	131.0 (63.5)	122.2 (55.8)	130.8 (58.4)	121.1 (53.0)
	FOR	741.2 (126.8)	740.3 (125.0)	737.9 (124.3)	744.4 (128.3)
P_max_ (N/cm²)	HEE	32.7 (6.7)	34.5 (6.1)	33.1 (7.0)	35.5 (6.5)
	MID	14.1 (6.2)	12.8 (5.7)	14.6 (5.8)	13.7 (6.5)
	FOR	35.8 (6.0)	35.7 (5.0)	35.1 (5.6)	35.5 (6.1)

NOTE: FLL = Fatigued lower limb; NFLL = Non-fatigued lower limb; T_c_ = Contact time; GL = Gait line; T_h–f_ = Time change Heel to Forefoot; F_max_ = Maximum force; P_max_ = Maximum pressure; HEE = Heel; MID = Midfoot; FOR = Forefoot.

## Data Availability

The data presented in this study are available on request from the corresponding author.

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
