# Peer review of "Fatigue-Induced Inter-Limb Asymmetries in Strength of the Hip Stabilizers, Postural Control and Gait Following a Unilateral Countermovement Vertical Jump Protocol"

_sports, 2021, doi:10.3390/sports9030033_

Round 1
Reviewer 1 Report
The authors present an interesting study that examines the effects of fatigue from unilateral jumping on a variety of outcomes. The research design is sound and on the whole the paper is well-written, but there are a few areas where the paper could be improved.
English - Presumably English is a second language for the authors, and on that basis you have done an excellent job in preparing the paper. However, there are instances throughout where the grammar or choice of word are not quite right. Getting someone with English as a first language to run a final check would help. I will upload a file with some examples highlighted, but the whole paper should be checked. Linked to this, there is a tendency to write very long sentences and those are where the flow of writing and English often break down.
Methods - you state n = 24 in the participants section but n = 23 in the stats analysis
Is the reliability of your measures known? Reliability should be confirmed
Results - Tables 2 and 3 need units adding
Discussion - The main finding is that CoP VL is altered with fatigue. However, it is not clear whether an increase in VL is desirable or undesirable. VL significantly increased in the fatigued limb and it needs to be clear whether that is a positive or negative outcome. If interlimb differences appear with fatigue simply as a result of a positive change in VL then that needs to be reflected in the discussion.
Conclusion - It may be obvious, but I think it is pertinent to note that fatigue is task specific, hence the different observations in the different tasks. The jump tests clearly caused fatigue and there may need to more investigation of changes in the control of jump performance (e.g. changes in dynamic stability during rebounding).

Reviewer 2 Report
This manuscript investigates the acute effects repeated unilateral vertical jumps on bilateral function and hip strength. While I have found this research interesting, I was wondering why untrained subjects were included in this project. Given that we already know that asymmetries in limbs can lead to injuries, shouldn't it make sense to conduct this project on elite or at least rained athletes?
More specific comment on the manuscript is outlined below.
Introduction
Line 36-44 I don't think it's necessary to keep this in this section. Perhaps, it con be moved to the discussion section.
What's the novelty of this project?
How exactly health care professional can benefit from this study?
